# Phenotypic and Genotypic Antimicrobial Susceptibility Testing of *Chlamydia trachomatis* Isolates from Patients with Persistent or Clinical Treatment Failure in Spain

**DOI:** 10.3390/antibiotics12060975

**Published:** 2023-05-28

**Authors:** Laura Villa, José Antonio Boga, Luis Otero, Fernando Vazquez, Ana Milagro, Paula Salmerón, Martí Vall-Mayans, María Dolores Maciá, Samuel Bernal, Luis Piñeiro

**Affiliations:** 1Microbiology Department, Central University Hospital of Asturias and Health Research Institute of Asturias (ISPA), 33011 Oviedo, Spain; joseantonio.boga@sespa.es (J.A.B.); opsklins@me.com (F.V.); 2Sexually Transmitted Infections Study Group of the Infectious Diseases and Clinical Microbiology Spanish Society (GEITS-SEIMC), 28003 Madrid, Spain; luis.otero.guerra@gmail.com (L.O.); anamilagro@gmail.com (A.M.); pausalmen@gmail.com (P.S.); mvall@lluita.org (M.V.-M.); mariad.macia@ssib.es (M.D.M.); samuel.bernal.sspa@juntadeandalucia.es (S.B.); luisdario.pineirovazquez@osakidetza.eus (L.P.); 3Microbiology Department, Cabueñes University Hospital, and Health Research Institute of Asturias (ISPA), 33394 Gijón, Spain; 4Department of Functional Biology, Microbiology Area, Faculty of Medicine, University of Oviedo, 33003 Oviedo, Spain; 5Microbiology Department, Miguel Servet University Hospital, 50009 Zaragoza, Spain; 6Microbiology Department, Vall d’Hebrón University Hospital, 08035 Barcelona, Spain; 7Vall’Hebron-Drassanes STI Unit, Infectious Diseases, Vall d’Hebrón University Hospital, 08035 Barcelona, Spain; 8Microbiology Department, Son Espases University Hospital, 07120 Palma de Mallorca, Spain; 9Infectious Diseases and Microbiology Unit, Valme University Hospital, 41014 Seville, Spain; 10Microbiology Department, Donostia University Hospital-Biodonostia Health Research Institute, 20014 San Sebastian, Spain

**Keywords:** *Chlamydia trachomatis*, antimicrobial susceptibility, treatment failure, sexually transmitted infection

## Abstract

The aim of this multicentre project (seven hospitals across the Spanish National Health Service) was to study the phenotypic and genotypic susceptibility of *C. trachomatis* to the main antimicrobials used (macrolides, doxycycline, and quinolones) in isolates from patients with clinical treatment failure in whom reinfection had been ruled out. During 2018–2019, 73 clinical isolates were selected. Sixty-nine clinical specimens were inoculated onto confluent McCoy cell monolayers for phenotypic susceptibility testing. The minimum inhibitory concentration for azithromycin and doxycycline was defined as the lowest concentration associated with an at least 95% reduction in inclusion-forming units after one passage in the presence of the antibiotic compared to the initial inoculum for each strain (control). Sequencing analysis was performed for the genotypic detection of resistance to macrolides, analysing mutations in the *23S rRNA* gene (at positions 2057, 2058, 2059, and 2611), and quinolones, analysing a fragment of the *gyrA* gene, and searching for the G248T mutation (*Ser83*->*Ile*). For tetracyclines, in-house RT-PCR was used to test for the *tet*(*C*) gene. The phenotypic susceptibility testing was successful for 10 isolates. All the isolates had minimum inhibitory concentrations for azithromycin ≤ 0.125 mg/L and for doxycycline ≤ 0.064 mg/L and were considered sensitive. Of the 73 strains studied, no mutations were found at positions T2611C or G248T of the *gyrA* gene. We successfully sequenced 66 isolates. No macrolide resistance-associated mutations were found at positions 2057, 2058, 2059, or T2611C. None of the isolates carried the *tet*(*C*) gene. We found no evidence for genomic resistance in this large, clinically relevant dataset.

## 1. Introduction

Of the numerous bacterial sexually transmitted infections (STIs), at the worldwide level, *Chlamydia trachomatis* is the most prevalent [1]. While the majority of *Chlamydia* infections are relatively easily and cheaply treated, in some cases infection results in serious complications that significantly increase treatment costs as well as morbidity, namely ectopic pregnancy, salpingitis, epididymitis, and ultimately infertility [2]. The fact that most infections are asymptomatic (as many as 70% of cases in women and 50% in men) [3,4] complicates both diagnosis and subsequent treatment and clearly points to the need for screening programmes—for high-risk groups in particular—in order to ensure long-term consequences are avoided or at least minimised [5,6].

The use of either tetracyclines or macrolides is the current gold standard for first-line treatment for *C. trachomatis*. Both work in the same way in that they bind to ribosomal subunits, to 30S in the case of tetracyclines and 50S in the case of macrolides [7]. Although antibiotic treatment with macrolides, tetracyclines, and eventually fluoroquinolones has been used successfully [8], clinical treatment failure rates range from 5 to 23%, depending on the population tested [9]. Many reasons for these treatment failures have been identified, and Pitts et al. [10] include the following in their list: “re-infection from a new or untreated partner, non-adherence to the treatment regimen, inadequate exposure to the antimicrobial because of host pharmacokinetics or short duration of treatment, and heterotypic or homotypic antimicrobial resistance” [10,11,12] (pp. 680–681). There is also both clinical and laboratory evidence for the reduced susceptibility and even resistance of *C. trachomatis* to these treatments, albeit in only a small number of patients [13,14,15,16,17]. In contrast to standard procedures in bacteriology, establishing the phenotypic antimicrobial susceptibility of different strains of *C. trachomatis* involves demonstrating that each can (or cannot) reproduce inside the cell [18]. While there are no universally agreed testing methods, and despite the fact that all the methods are time-consuming and complex at the technical level and therefore need to be carried out in specialised labs [19], the most common system employed to ascertain *C. trachomatis* susceptibility is based on using cell cultures and adding serial dilutions of different antibiotics [20]. Strains found to be resistant can subsequently be analysed with molecular techniques to ascertain potential genetic markers of resistance. Moreover, some genetic mutations have been associated with antibiotic resistance, and these could be analysed directly from the sample [21,22,23].

Patients with recurrent *C. trachomatis* infection present a particular challenge in treatment terms as their infections tend to show macrolide resistance [9,15,24]. This has been linked to mutations at positions 2057, 2058, 2059, and 2611 (following the *E. coli* numbering system), which coincide with the peptidyl transferase region of the *23S rRNA* [22,25].

Furthermore, in terms of *Chlamydia* strains that affect swine, frequent resistance to tetracyclines was reported for *Chlamydia suis*, which replicates in the same type of host cell, which raises the possibility of the horizontal transfer of genetic resistance between the two species [26]. In another study, resistant isolates were found to have similar genetic characteristics: foreign genomic islands (between 6 and 13.5 kb) in the chlamydial chromosome containing genes that encode antibiotic efflux pump (*tet*(*C*)) and regulatory repressor (*tetR*), a unique insertion sequence (IScs605), as well as up to ten genes that have a role in replicating and mobilising the plasmid [21].

As regards fluroquinolones, the second-line treatment for *C. trachomatis* infections [27], strains can develop resistance in vitro when subjected to subinhibitory concentrations of the drug [8,28]. There is evidence to suggest that such resistance is conferred by a point mutation in the region that determines *gyrA* quinolone resistance (QRDR), whereby isoleucine is substituted by serine at amino acid position 83 (S83I, according to *E. coli* numbering) in the corresponding protein [29,30,31,32]. Recent studies have identified the analogous QRDR of similar point mutations in *parC* that result in fluoroquinolone resistance [33,34,35]. In addition, other mechanisms of resistance to fluroquinolones (such as drug efflux modification or drug permeation) may contribute to the resistance pattern [36].

The objective of this multicenter study was to analyse the phenotypic and genotypic susceptibility of *C. trachomatis* to the main antimicrobials used in its treatment (macrolides, tetracyclines, and quinolones), isolated from samples from patients with persistent or clinical treatment failure. This was the first such study carried out in Spain.

## 2. Results

Isolates were retrieved from 73 patients with persistent *C. trachomatis* infections. The clinical specimens for persistently infected patients were from 37 men (29 urethral swabs, 4 rectal swabs, 3 urine samples, and 1 semen sample) and 36 women (9 cervical swabs, 25 vaginal swabs, 1 pharyngeal swab, and 1 rectal swab). Regarding the antibiotic treatment received, 50 (68.5%) patients had been treated with azithromycin (26 women, 24 men), 13 (17.8%) with doxycycline (6 women, 7 men), 9 (12.3%) with azithromycin plus doxycycline (4 women, 5 men), and 1 (1.4%) male patient with levofloxacin. The clinical and epidemiological data are summarised in Appendix A.

### 2.1. Isolate Retrieval

#### 2.1.1. MIC Determination

The minimum inhibitory concentrations of azithromycin and doxycycline were determined. Phenotypic susceptibility testing was completed for 10 isolates, measuring MICs for azithromycin and doxycycline. All the isolates had MICs for azithromycin ≤ 0.125 mg/L and for doxycycline ≤ 0.064 mg/L and were considered sensitive to these antibiotics (Table 1). The phenotypic analysis of fluoroquinolone susceptibility was not performed because the inoculum was insufficient to carry out the MICs.

#### 2.1.2. SNP Genotyping

PCR-based genotyping of single nucleotide polymorphisms (SNPs) was performed to detect resistance-associated mutations. Specifically, the T2611C mutation in the *23S rRNA* and the G248T mutation in the *gyrA* genes were detected. In the 73 strains analysed using SNPs genotyping, no mutations were found at position T2611C, which confers resistance to macrolides, or position G248T of the *gyrA* gene, which confers resistance to quinolones (Table 1).

*23S-rRNA* gene sequencing analysis was performed for the genotypic detection of resistance to macrolides, analysing mutations in the two copies of the *23S rRNA* gene (at positions 2057, 2058, 2059, and 2611). Of the 73 isolates, 66 were successfully sequenced. However, in seven strains, there was a sequencing failure. The amplicons were compared with the reference strain D/UW-3/CX in the GenBank database (NC000117.1). No resistance-associated mutations were found at 2057, 2058, 2059, or T2611C (*E. coli* numbering) (Table 1).

#### 2.1.3. *tet*(*C*) Gene Detection

In tetracyclines, an “in-house” RT-PCR was used to test for the presence of the *tet*(*C*) gene. None of the isolates carried the *tet*(*C*) gene, which confers resistance to tetracyclines (Table 1).

### 2.2. Genetic Characterization

Genotype E was the most prevalent (29/67, 43.3%), followed by genotypes F, D, and G (22.4%, 13.4%, and 10.4%, respectively). MLST was performed in 17 samples and detected 10 different STs across 5 genotypes, indicating that this technique has a better discriminatory capacity than *ompA* genotyping (Appendix A).

## 3. Methods

### 3.1. Patient Recruitment

During 2018 and 2019, patients were recruited in seven tertiary hospitals in the Spanish National Health Service network (Central University Hospital of Asturias, Oviedo; Cabueñes University Hospital, Gijón; Miguel Servet University Hospital, Zaragoza; Vall d’Hebrón University Hospital, Barcelona; Son Espases University Hospital, Palma de Mallorca; Valme Univeristy Hospital, Seville and Donostia University Hospital, San Sebastián). The participating hospitals are responsible for the microbiological diagnosis of STIs in their corresponding health regions, with most patients being seen in STI clinics, emergency departments, gynaecology units, or by family physicians.

Patients were deemed to have a persistent infection if they had tested positive at least twice using a *C. trachomatis*-specific assay (a nucleic acid amplification test with an amplification cycle < 35) and had fully adhered to the prescribed treatment regimens and management of the infection in line with current guidelines (including testing of sexual partners, abstinence, or protected sex for 1 week after a single dose or until the completion of a longer course of treatment, and a test of cure after >3 weeks) [27]. Samples from patients with clinical treatment failure in which *C. trachomatis* was detected were stored at −80 °C. Reinfection was ruled out based on medical record review and genotyping of the samples.

The specimens were sent to the Microbiology Service at the Central University Hospital of Asturias for phenotypic and genotypic antimicrobial susceptibility testing and to the Microbiology Service at Donostia University Hospital for molecular characterization.

### 3.2. Culture Methods

#### 3.2.1. Stock Inoculum Culture

Clinical specimens were inoculated onto confluent McCoy cell monolayers in culture tubes. Culture tubes were incubated for 90 min in a laminar flow hood, and after this time, the following were added: 1.5 mL of minimal essential medium (MEM, Gibco, Hemel Hempstead, UK) supplemented with 10% fetal bovine serum and 2 mM L glutamine (Sigma-Aldrich, Gillingham, UK), 1 mg/L cycloheximide (Sigma-Aldrich), 100 mg/L of gentamicin (Gibco), 25 U/mL fluconazole (Sigma-Aldrich), and 100 mg/L vancomycin (Sigma-Aldrich). Subsequently, the culture was incubated in an incubator at 37 °C with 5% CO_2_ on an inclined rack for 5 days to produce a stock inoculum of each strain for antibiotic susceptibility testing assays.

To detect the presence of *C. trachomatis*, direct immunofluorescence was performed with specific monoclonal antibodies against *C. trachomatis* conjugated with fluorescein (Vircell, Granada, Spain). First, the monolayers were scraped into the medium in the culture tubes, and 500 μL was reserved at −80 °C for antibiotic susceptibility testing assays or new cultures. The rest of the tube content was processed for immunofluorescence. For this, 5 mL of PBS was added to the scraped tube, and it was centrifuged for 10 min at 1800 rpm. The supernatant was decanted, and the sediment was resuspended in the remaining liquid. With a pipette, this concentrated cell suspension was aspirated and deposited in the wells of a slide. Once dry, it was fixed with formaldehyde.

#### 3.2.2. Susceptibility Assays

Minimum inhibitory concentrations (MICs) of azithromycin and doxycycline were determined using the method described by Pitt et al. [10]. The test was carried out in a culture chamber (Nunc™ Lab-Tek™ glass chamber slide system, Thermo Fisher Scientific, Rochester, NY, USA). When the chambers had a confluent monolayer, the medium was decanted, and the chambers were inoculated with 250 μL of the aforementioned inoculum. They were centrifuged for 1 h at 1350 g to facilitate infection. Serial solutions (1:2) of antibiotic (0.125–2 mg/L azithromycin or 0.064–1 mg/L doxycycline) were added to the supplemented MEM. Three controls were included in each chamber to identify assay failure: one well with maintenance medium, another with maintenance medium with the antibiotic to be studied (first dilution), and a third with maintenance medium free of the antibiotic to be studied but with *C. trachomatis* inoculum. The chambers were incubated for 48 h at 37 °C in 5% CO_2_.

To assign the MICs of each antibiotic to each strain, the aforementioned monoclonal antibody staining method (*Chlamydia trachomatis* MAb, Vircell) was used directly on the chambers for subsequent visualisation under the fluorescence microscope. As described by Storm et al. [37], the MIC was assigned to the lowest antimicrobial concentration associated with an at least 95% reduction in inclusion-forming units after one passage in the presence of the antibiotic compared to the initial inoculum for each strain (control).

### 3.3. Genotypic Methods

#### 3.3.1. DNA Extraction

The nucleic acid extraction step was carried out using a Nimbus automated liquid handling workstation (Hamilton, Reno, NV, USA).

#### 3.3.2. Analysis of Macrolide and Quinolone Resistance-Associated Mutations by PCR-Based Genotyping of Single Nucleotide Polymorphisms

PCR-based genotyping of single nucleotide polymorphisms (SNPs) was performed to detect resistance-associated mutations. Specifically, T2611C and G248T mutations in the *23S rRNA* and *gyrA* genes were detected using Brilliant III Ultra-Fast QPCR Master Mix (Agilent Technologies, Santa Clara, CA, USA) supplemented with primers (900 nM) and probes (250 nM) described in Table 2. A 168-mer DNA oligonucleotide was used as a mutant control (Table 2). Reactions were run on a CFX 96 real-time PCR system (Bio-Rad, Hercules, CA, USA). The cycling conditions consisted of an initial denaturation and Taq activation step at 95 °C for 10 min, followed by 40 cycles consisting of denaturation at 95 °C for 15 s, annealing at 62 °C for 45 s, and extension at 50 °C for 30 s.

#### 3.3.3. Analysis of Macrolide Resistance-Associated Mutations by Sequencing

Mutations at positions 2057, 2058, 2059, and 2611 in the peptidyl transferase region of *23S rRNA* were detected by sequencing the two copies of the *23S rRNA* gene using the primers listed in Table 1 and following the protocol described by Misyurina et al. [22]. The amplicons were sequenced and compared with the reference strain D/UW-3/CX in the GenBank database (NC000117.1) using MEGA 3.1.

#### 3.3.4. Analysis of Tetracycline Resistance by Detecting the Presence of the *tet*(*C*) Gene

For tetracyclines, an “in-house” RT-PCR was used to test for the presence of the *tet*(*C*) gene. PCR was carried out in a total volume of 25 μL, with 5 μL of dNTP (4 μM), 0.50 μL of MgCl_2_ (25 μM), 0.10 μL of Taq polymerase (5 U/μL), 2.5 μL of buffer (10X) (Bioline Reagents, London, UK), 1 μL of forward and reverse primers (12.5 μM) (Table 2) as indicated, and 5 μL of a purified DNA sample. Amplification was done in a CFX96 real-time PCR detection system (Bio-Rad, Hercules, CA, USA), under the following conditions (as per [15]): 94 °C for 5 min, followed by 40 cycles of 94 °C for 1 min, 55 °C for 1 min, and 72 °C for 1:30 min, and a final extension at 72 °C for 10 s. For the nested PCR reaction, 3 μL of the first-round PCR product was added to 22 μL of a reaction mix prepared as described above except with the substitution of the primer pair. A 1.2% agarose gel was used to analyse PCR products, which were stained with ethidium bromide to confirm amplification using a115 pb fragment of DNA.

#### 3.3.5. Genetic Characterization

For *ompA* genotyping, the 990-bp fragment of interest was amplified using a conventional PCR system [38], resulting in a Sanger sequencing of the amplicons (3130XL Genetic Analyzer, Applied-Biosystems, Foster City, CA, USA), which was then analysed using the Basic Local Alignment Search Tool (http://www.ncbi.nlm.nih.gov/blast/Blast.cgi, accessed on 22 november 2022). Additionally, in randomly selected samples, sequence typing was performed using multilocus sequence typing (MLST), whereby five highly variable genes (*hctB*, CT058, CT144, CT172, and *pbpB*) were amplified and then bidirectionally sequenced [39].

## 4. Discussion

Although bacterial resistance to antibiotics is a major challenge currently, in *C. trachomatis* it is actually extremely rare. Indeed, the potential of this bacterial species to develop antimicrobial resistance has been little studied. That said, treatment failure in *C. trachomatis* infections despite the use of recommended therapeutic antimicrobials has been reported [14,15,40,41], and there have been isolated reports of genetically conferred resistance [22,42], as has the fact that resistance can easily be selected for in vitro by exposure to subinhibitory antimicrobial concentrations [8,19,23,43]. On the other hand, no *C. trachomatis* strains that demonstrate stable resistance to the traditional and typical antimicrobial agents used in therapy have been isolated, and neither have the mechanisms of putative antimicrobial resistance been described for isolates obtained from patients with treatment failure. It must be acknowledged that little has been clarified in terms of the prevalence, if any, of circulating resistance traits [19]. The first-line drugs for the treatment of *C. trachomatis* infection are macrolides [44]. Fohner et al. [45] describe how the mechanism of action revolves around their ability to bind to the bacterial 50S ribosomal subunit; this binding causes cessation of bacterial protein synthesis and, in turn, bacterial growth.

Binet and Maurelli designed an in vitro model that described a *C. trachomatis* serovar L2 population known to have a four-fold lower susceptibility to josamycin and spiramycinas, as well as a susceptibility that was eight times less than azithromycin and erythromycin, which they found to be the result of mutations in the *rplD* gene coding for the L4 ribosomal protein. When there was no antibiotic present, the growth of this population was reduced, its inclusions were smaller, and the number of infectious particles produced was smaller. These findings suggest that, in vivo, compensatory mutations are essential for chlamydial resistance [46].

In clinical isolates of *C. trachomatis* resistant to erythromycin, azithromycin, and josamycin, Misyurina et al. [22] described mutations at A2058C and T2611C (*E. coli* numbering) in the peptidyl transferase region of *23S rRNA* genes. They also found a triple mutation in a non-conserved region of the protein L22 (i.e., Gly52 (GGC)3Ser(AGC), Arg65(CGT)3Cys(TGT), and Val77(GTC)3Ala(GCC)) [22]. While the exact role played by these amino acid replacements in terms of *C. trachomatis* antibiotic resistance has yet to be fully elucidated, it is assumed that they are compensatory mutations that help to maintain virulence in the chlamydial strains where they are present.

In our study, phenotypic susceptibility testing was completed on 10/69 isolates from clinical treatment failures. All the isolates had a MIC ≤ 0.125 mg/L for azithromycin, and hence, they were considered sensitive to this antibiotic. In the 73 strains analysed by SNP genotyping, no mutations were found at position T2611C that confer resistance to macrolides. Seeking to detect other mutations described (A2058C, etc.), 66 of the 73 isolates were successfully sequenced with primers specific for the two copies of the *23S rRNA* gene [22]; however, no resistance-associated mutations were found at 2057, 2058, 2059, or T2611C (*E. coli* numbering).

Our results accord with those of Hadfield et al. [47], who analysed 563 full genomes, 455 of them novel, of isolates collected between 1957 and 2012 in clinical samples from a wide range of countries (n = 21) across Europe, North America, South America, Africa, Asia, and Australia. Included in their analyses was the *23S rRNA* gene, known to be involved in bestowing resistance to the macrolide azithromycin. It should be noted here that the samples used by Hatfield et al. were in fact from patients undergoing routine diagnostic procedures and not from patients in whom treatment had failed. Despite none of the isolates in Hadfield et al.’s study actually being antibiotic-resistant, they systematically searched for already known resistance alleles, either fixed or heterozygous, in a circulating population, finding no evidence of genomic resistance. A more plausible, explanation was put forward by Jiang et al. [42] (one that has indeed been shown to play an important role in macrolide resistance), namely that the mechanisms underlying antibiotic resistance in *Chlamydiae* also impact strongly on infectivity, the result being that the potential for the emergence in vivo of highly resistant clones of *Chlamydiae* is greatly reduced.

The results of both a meta-analysis [48] and a Cochrane systematic review [49] evaluating randomised clinical trials of treatment for urogenital chlamydia infection in men found higher rates of treatment failure when azithromycin as opposed to doxycycline was used. Doxycycline has also been found to be more effective for rectal *C. trachomatis* infection than azithromycin, both in male and female patients [48,50]. A randomised trial examining the treatment of rectal *Chlamydia* infection among men who have sex with men reported a 100% clear-up rate with doxycycline and 74% with azithromycin [51]. What is more, a review of women with *C. trachomatis* found that the bacteria was also detected in the anorectal region in 33–83% of such women, and, interestingly, this was not linked to reports of receptive anorectal sexual activity [52]. In our study, 68.5% of patients had been treated with azithromycin, 17.8% with doxycycline, 12.3% with azithromycin plus doxycycline, and 1.4% with levofloxacin. This may be one of the factors underlying the treatment failure, but the finding should be interpreted with caution due to the relatively small number of patients.

Doxycycline is a semisynthetic tetracycline and a first-line treatment for *C. trachomatis*, especially when the infection is rectal and where LGV strains are involved [27]. Resistance to tetracycline antibiotics in *C. trachomatis* involves mechanisms similar to those reported for the closely related and highly recombinogenic species *C. suis* [19,20,21] in that it involves a genomic island carrying a *tet*(*C*) allele. This provides evidence for the horizontal transfer of antibiotic resistance genes [19], despite the fact that the literature does not currently provide evidence for stable tetracycline resistance in clinical strains of *C. trachomatis*. The strains involved in the numerous reports of patients with apparent treatment failure generally have a heterotypic resistance pattern whereby only a small proportion of the bacterial population survives after antibiotic treatment [13,15]. In our study, the MIC to doxycycline in the 10 isolates phenotypically analysed was ≤0.064 mg/L, and none of the isolates carried the *tet*(*C*) gene.

Nevertheless, Pitt et al. [10], studying antimicrobial susceptibility in isolates from patients with either persistent or successfully treated *C. trachomatis* infection, found higher doxycycline MICs in persistently infected patients. However, it is still unclear what the underlying cause of this shift is and what impact it may have on clinical outcomes. As doxycycline is currently the preferred first-line therapy for non-gonococcal urethritis—*C. trachomatis* being the most common pathogen [12]—it is clearly important to understand the significance of these elevated doxycycline MICs in persistently infected patients. This becomes an even more relevant line of research since there is a move towards doxycycline being given as pre-exposure prophylaxis to men who have sex with men [53]. These data may strengthen the case for recommending treatment for all patient groups.

Regarding quinolone resistance, in our study, we did not find G248T mutations in *gyrA* QRDR (S83I in the corresponding protein, *E. coli* numbering). Our results are in agreement with those of Yokoi et al. [54], who screened for but did not find fluoroquinolone resistance-associated mutations in the *gyrA* and *parC* genes of *C. trachomatis* following levofloxacin treatment.

Patients may still test positive for *Chlamydia* following treatment for a variety of reasons, one of which is whether they exhibit homotypic or heterotypic antimicrobial resistance [9]. Homotypic resistance, in which most of the organisms survive at concentrations well above the MIC, is genetically inherited and has not yet been documented in *C. trachomatis* [55]. Heterotypic resistance, or phenotypic switching, results from a heterogeneous population (less than 1%) comprising resistant and susceptible organisms that replicate the pattern whereby small numbers of organisms survive antimicrobial concentrations above the MIC [20]. Rather than being a genetically inherited trait, this is the result of the bacteria adapting and becoming less susceptible to the antimicrobial, i.e., forms that, in the presence of antibiotics, are slow-growing, non-reproductive, or persistent revert to replicating forms once the antibiotic is withdrawn, leading to the reactivation of the infection [10]. This phenomenon of small numbers of chlamydial organisms surviving even when high levels of antimicrobials are administered may well have evolved because of selective pressure in the face of repeated exposure to antimicrobials, although it may equally be an innate characteristic of certain isolates that are able to establish latent infection [15,20,56]. Patients with high bacterial loads of *C. trachomatis*, such as those with urethritis, are known to exhibit heterotypic resistance [57], although, as has been noted already, clinically significant phenotypic reduced susceptibility to antimicrobials has rarely been reported in *C. trachomatis* [12,58]. It could be that this small proportion of strains expressing heterotypic resistance would have lower fitness or clonal dissemination. However, on the other hand, the distribution of *C. trachomatis* genotypes in these strains of patients with treatment failure in the present study was similar to that described in the general population without treatment failure in the same period in Spain [59]. Genotype E was the most prevalent (43.3%), followed by genotypes F, D, and G (22.4%, 13.4%, and 10.4%, respectively).

*C. trachomatis*’s unique developmental cycle may also explain its apparent lack of antimicrobial resistance. Gene replication of *C. trachomatis* occurs inside an intracellular inclusion in an infected epithelial cell. This isolation would, of course, make the acquisition of antibiotic resistance genes from other organisms difficult [20]. Nonetheless, in vitro resistance has been demonstrated following selective pressure from exposure to subinhibitory concentrations of antimicrobials, for example, fluoroquinolones [8]. Finally, the persistence of the infection for other metabolic reasons could be another cause of treatment failure [60].

Our study has limitations, such as the small number of strains for which phenotypic antimicrobial susceptibility testing was completed. This is attributable to the culture of *C. trachomatis* having a low yield and the technique to assess phenotypic susceptibility being laborious, which is a common problem; the number of strains analysed is also small in other published studies [10,22,23,28,42]. Further, in the genotypic study, not all known mutations associated with antibiotic resistance were studied; rather, we covered the most frequently described mutations. We believe that the development of whole-genome sequencing using strategies that improve its cost-effectiveness in difficult-to-grow bacteria such as *C. trachomatis* will make it more feasible to analyse the complete genome in strains of infections with treatment failure to seek evidence for antimicrobial resistance [61].

## 5. Conclusions

Our study, despite starting from a selected population with clinical failure, in which possible reinfections were initially ruled out with the methods used, did not detect phenotypic or genotypic resistance to the antibiotics tested (azithromycin, doxycycline, and quinolones) in *C. trachomatis.* On the other hand, the antibiotic susceptibility analysis allowed the standardisation of protocols for useful molecular techniques to be applied in future direct samples of *C. trachomatis* infections with suspected clinical failure, especially for macrolides. The preliminary results of this study should be extended. In addition, other possible causes of these failures and mechanisms that hinder antibiotic activity should be investigated.

## Figures and Tables

**Table 1 antibiotics-12-00975-t001:** Results of phenotypic and genotypic antimicrobial susceptibility testing in macrolides, tetracyclines, and quinolones.

Antibiotic Susceptibility							
Macrolides					Tetracyclines		Quinolones
**Phenotypic (MIC)**	Genotypic (n = 66)				Phenotypic (MIC)	Genotypic	Genotypic
**Azithromycin (mg/L) (n = 10)**	T2611C	A2057G	A2058C	A2059G	Doxycycline (mg/L) (n = 10)	*tet*(*C*) gene (n = 73)	G248T
**≤0.125**	ND	ND	ND	ND	≤0.064	ND	ND

ND: not detected; MIC: Minimum inhibitory concentrations.

**Table 2 antibiotics-12-00975-t002:** Nucleotide sequences of primers and probes used for detection of resistance-associated mutations by PCR-based genotyping of single nucleotide polymorphisms, sequencing, and detection of the presence of the *tet*(*C*) gene.

Antibiotic Resistance	Technique	Target	Function	Name	Sequence (5′-3′) ^1^
Macrolides	SNP	*23S rRNA*	Forward primer	ClaMacro-F	GTTCATATCGACGTGGCGGT
		(T2611C)	Reverse primer	ClaMacro-R	GTATCCTGCGCCCACGAA
			Probe Wild Type	CT-2611-S-VIC	CAGTTTGGTCTCTATC
			Probe Mutant	CT-2611-R-FAM	CAGTTTGGTCCCTATC
			Control Mutant	CCC-Cla-Macro	GAGTTCATATCGACGTGGCGGTTTGGCACCTCGATGTCGGCTCATCGCATCCTGGGGCTGGAGAAGGTCCCAAGGGTTTGGCTGTTCGCCAATTAAAGCGGTACGCGAGCTGGGTTCAAAACGTCGTGAGACAGTTTGGTCCCTATCCTTCGTGGGCGCAGGATACTT
Quinolones	SNP	*gyrA*	Forward primer	Quino-CT-F	TTTGCGGTGATACTTCCGG
		(G248T)	Reverse primer	Quino-CT-R	CCCAATCCTGTGCCATCC
		(S83I)	Probe Wild Type	Qui-WT-VIC	ATGGAGAAAGTGTCATTT
			Probe Mutant	Qui-MUT-FAM	GGAGAAAATGTCATTTAT
			Control Mutant	AAT-Qui-Ctrl	TTTGCGGTGATACTTCCGGAGATTATCACCCCCATGGAGAAAATGTCATTTATCCTACTTTAGTAAGGATGGCACAGGATTGGG
Macrolides	Sequencing	*23S rRNA*	Forward primer	rr-f	AAGTTCCGACCTGCACGAATGG
			Reverse primer	rr-r	TCCATTCCGGTCCTCTCGTAC
			Forward primer	rrg-f	AATTCCTTGTCGGGTAAGTTC
			Reverse primer	al1-r	CGTTATGATCCCAGGATCCCT
			Reverse primer	al2-r	CCCAATATAGAACCGAAAATTCGA
Tetracyclines	RT-PCR	*tet*(*C*)	Forward primer	CT-tetS	AGCACTGTCCGACCGCTT
			Reverse primer	CT-tetA	TCCGGCGTAGAGGATCCA

## Data Availability

Data on clinical/epidemiological characteristics and microbiological information were recorded in a database, which was anonymised for subsequent analysis.

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
