# Peer review of "Phenotypic and Genotypic Antimicrobial Susceptibility Testing of *Chlamydia trachomatis* Isolates from Patients with Persistent or Clinical Treatment Failure in Spain"

_antibiotics, 2023, doi:10.3390/antibiotics12060975_

Round 1
Reviewer 1 Report
1. The title of this manuscript is “phenotypic and genotypic antimicrobial susceptibility testing…”, but the authors mainly focused on the sequence of the 23S-rRNA gene (at positions 2057, 2058, 2059, and 2611) and, analyzing a fragment of the gyrA gene, searching for the G248T mutation (Ser83->Ile). Is there any direct and strong evidence to support that antimicrobial susceptibility is related to the SNP of these gene sites? The detection of 23 S rRNA and gyrA cannot represent the genotype, because many other possibilities lead to phenotype change.
2. In the section of “Patient recruitment”, the content “with a total catchment population of over 3 million, including approximately 2 million 15- to 65-year-olds.” is not related to this study so it can be deleted.
3. If the mutation of the relevant gene causes drug resistance, why did not knock out or replace the gene to get the mutant to support the description? Why are there 7 strains of sequencing failure, the proportion of SNP genotyping used is in 73 strains, but the subsequent mutation is based on the results of 69 strains, should the subsequent proportion of 69 strains prevail.
4. The description of the Results section is too less.
5. The format of article writing should be uniform. Line 256 and other first-line spaces are different.
6. in lines 136 and 159, it should be “CO2” instead of “CO2”.
Minor editing of English language required
Author Response
Dear Reviewer 1,
Thank you very much for the comments and suggestions, I think they add great value to the article. I have carefully reviewed your suggestions for improving the manuscript, and I am sending you the new document in which I have made corrections for your review. I will try to respond to your comments on the changes to be made.
- The title of this manuscript is “phenotypic and genotypic antimicrobial susceptibility testing…”, but the authors mainly focused on the sequence of the 23S-rRNA gene (at positions 2057, 2058, 2059, and 2611) and, analyzing a fragment of the gyrA gene, searching for the G248T mutation (Ser83->Ile). Is there any direct and strong evidence to support that antimicrobial susceptibility is related to the SNP of these gene sites? The detection of 23 S rRNA and gyrA cannot represent the genotype, because many other possibilities lead to phenotype change.
In the title when referring to "phenotypic and genotypic antimicrobial susceptibility" we refer generically to the way of describing the susceptibility methods used in the study. We agree that the detection of 23 S rRNA and gyrA cannot represent the genotype, because many other possibilities lead to phenotype change, and that any genotypic method that does not use WGS does not analyze all the possibilities of genetic mutation, but in our study we have analyzed the most frequently described ones associated with antibiotic resistance in C. trachomatis.
Our work has limitations, in the genotypic study, not all known mutations associated with antibiotic resistance were studied, rather we covered the most frequently described mutations [21,22,25,29,32,42].
In the case of macrolides, PCR-based genotyping of single nucleotide polymorphisms (SNPs) was performed to detect resistance-associated mutations, specifically, T2611C. To do this, we base ourselves on the article published by Jiang et al. (Reference number 42), the T2611C mutation was the most frequent in the macrolide-resistant isolates, and may have been responsible for the selective resistance. The analysis of mutations at positions 2057, 2058, 2059 by PCR-based genotyping of single nucleotide polymorphisms, is very difficult, because the positions of nucleotide are close and we would not obtain results. For this reason, sequencing analysis was performed for the genotypic detection of resistance to macrolides, analysing mutations in the 23S rRNA gene (at positions 2057, 2058, 2059, and 2611).
In fluorquinolones, analyzing a fragment of the gyrA gene, searching for the G248T mutation with a PCR-based genotyping of SNP, studies suggest that this mutation is a common fluorquinolone resistance mechanism in C. trachomatis. Other studies have identified the analogous QRDR of similar point mutations in parC that result in fluoroquinolone resistance. We believe that future studies, looking for other mutations, would be necessary to complete this work.
- In the section of “Patient recruitment”, the content “with a total catchment population of over 3 million, including approximately 2 million 15- to 65-year-olds.” is not related to this study so it can be deleted.
In the section “Patient recruitment”, what we intend in this paragraph is to describe the population that corresponds to all the provinces that have collected samples. Although the selection of patients has been random, including those clinical treatment failure in which C. trachomatis was detected, and the incidence rate is not calculated. We agree with the reviewer and the paragraph has been removed.
- If the mutation of the relevant gene causes drug resistance, why did not knock out or replace the gene to get the mutant to support the description? Why are there 7 strains of sequencing failure, the proportion of SNP genotyping used is in 73 strains, but the subsequent mutation is based on the results of 69 strains, should the subsequent proportion of 69 strains prevail.
We agree with the reviewer and this study should be very interesting but our laboratory is a clinical microbiology laboratory and we lack the necessary technology to carry out the Knockout technology.
In all 73 isolates, both copies of the 23S rRNA peptidyl transferase region were sequenced to look for mutations at positions 2057, 2058, 2059, and 2611. Of these, 66 were sequenced successfully, but sequencing failed in 7 isolates. Although the amplicon was purified using Exo-Bap Mix (EURx, Gdnsk, Poland), and the sequencing was repeated, no legible sequence was obtained (Line235-236).
Isolates were retrieved from 73 patients with persistent C. trachomatis infections. Sixty-nine clinical specimens were inoculated onto confluent McCoy cell monolayers for phenotypic susceptibility testing, since 4 strains were DNA extracted. The phenotypic susceptibility testing was completed in 10 isolates. Therefore, the subsequent proportion of 69 strains must prevail (10/69). I have already corrected it in the text (Line 277).
- The description of the Results section is too less.
We agree that the results section is too less by comparison with the introduction and discussion sections. But these are the results we have obtained. Although we have not found any confirmation of the phenotypic or genotypic presence of resistance. I consider that the results obtained are valuable and indicate that other mechanisms responsible for treatment failure should be sought or modification of previously standardized procedures for the analysis of phenotypic resistance.
- The format of article writing should be uniform. Line 256 and other first-line spaces are different.
I have reviewed the article format and changed errors in the spaces.
- In lines 136 and 159, it should be “CO2” instead of “CO2”.
I have changed in lines 136 and 159 “CO2” for “CO2”.

Reviewer 2 Report
The subject of the manuscript is interesting, and despite the lack of confirmation of phenotypic or genotypic presence, the obtained results are valuable and indicate that other mechanisms responsible for treatment failure should be sought or the previously standardized procedures of phenotypic resistance analysis should be modified.
Major comments
Please explain why phenotypic analysis of fluoroquinolone susceptibility was not performed
Please explain what were the criteria for selecting only 10 strains for which the MIC for azithromycin and doxycycline was determined
Line 313: azithromycin is not a tetracycline, please verify this part of the discussion
Molecular characterization has been completely omitted from the discussion, please complete
Minor comments:
Line 56,260,269, 294 : in many places, complete the reference number with the quoted name or names of authors
Line 79-87: this part should be moved to discussion
Line 135: oven? More like “an incubator”
Chapter: susceptibility assays: please specify the wavelength
Line 262, 278 and others: please replace with "susceptibility"
Line 266, 297: please use italics (in vivo)
Line 339 - please use square brackets
Author Response
Dear Reviewer 2,
Thank you very much for the comments and suggestions, I think they add great value to the article.
The subject of the manuscript is interesting, and despite the lack of confirmation of phenotypic or genotypic presence, the obtained results are valuable and indicate that other mechanisms responsible for treatment failure should be sought or the previously standardized procedures of phenotypic resistance analysis should be modified.
Major comments
- Please explain why phenotypic analysis of fluoroquinolone susceptibility was not performed.
The phenotypic analysis of fluoroquinolone susceptibility was not performed due to the inoculum was insufficient to carry out the minimum inhibitory concentrations. Only Minimum inhibitory concentrations of azithromycin and doxycycline could be determined. I have added this phrase to the results section (Line 228-229).
- Please explain what were the criteria for selecting only 10 strains for which the MIC for azithromycin and doxycycline was determined.
Sixty-nine clinical samples were inoculated into confluent McCoy cell monolayers for phenotypic susceptibility testing; of these, we only made 10 C. trachomatis isolates viable for susceptibility testing. Although, chlamydial culture followed by immunofluorescence microscopic evaluation is the traditional gold standard method for the diagnosis of chlamydial infection and represents the definitive method for viability evaluation, allowing antimicrobial resistance studies. The culture of C. trachomatis having a low yield, the sensitivity has been affected by inadequate specimen collection, storage and transport, toxic substances in clinical specimens and overgrowth of cell cultures by commensal microbes.
- Line 313: azithromycin is not a tetracycline, please verify this part of the discussion
If it's a mistake, azithromycin is not a tetracycline, is a macrolide. I have already changed it in the discussion (line 316).
- Molecular characterization has been completely omitted from the discussion, please complete.
The molecular characterization is included in the discussion in Line 363-365. In this work, we wanted to relate the distribution of C. trachomatis genotypes in these strains of patients with treatment failure and compare it with the distribution of genotypes present in the general population without treatment failure in the same period in Spain (reference 59). The distribution of C. trachomatis genotypes was similar. The genotype E was the most prevalent (43.3%), followed by genotypes F, D, and G (22.4%, 13.4%, and 10.4%, respectively). This fact has been added to discussion section (lines 367-369).
Minor comments:
Line 56, 260, 269, 294: in many places, complete the reference number with the quoted name or names of authors
I have already completed the reference number with the quoted name or names of authors.
Line 79-87: this part should be moved to discussion
I have reviewed this part, and I think that is the introduction to the resistance to tetracyclines. I have summarized it and I have also changed the discussion.
Line 135: oven? More like “an incubator”
I have already replaced “oven” for “an incubator”.
Chapter: susceptibility assays: please specify the wavelength
Reactions for analysis of macrolide and quinolone resistance-associated mutations by PCR-based genotyping of single nucleotide polymorphisms were run on a CFX 96 real-time PCR system. The probes used were marqued with Fluorophores, FAM (Probe Mutant) and VIC (Probe Wild Type). The wavelength were FAM: 520 and VIC: 548.
Line 262, 278 and others: please replace with "susceptibility"
I have already replaced “sensitivity” with "susceptibility".
Line 266, 297: please use italics (in vivo)
I have already used italics (in vivo).
Line 339 - please use square brackets
I have already used square brackets.

Round 2
Reviewer 1 Report
1. First and foremost, the data in the results section is too less and the description is too short. It is suggested to expand this section.
2. In line 62, a space is missing before “In contrast to”.
3. In line 219, there is an additional comma in "(6 women, 7 men,)".
4. In line 380, please use italics for “C. trachomatis”.
5. I personally believe that the specific results of 'Susceptibility assessments' can be plotted and included in the paper to make the data clearer.
6. I personally think the summary of the last paragraph can be separated into the fifth part of the article.
Author Response
Dear Reviewer,
Thank you very much for the comments and suggestions, I think they add great value to the article. I have carefully reviewed your suggestions for improving the manuscript, and I am sending you the new document in which I have made corrections for your review. I will try to respond to your comments on the changes to be made.
- First and foremost, the data in the results section is too less and the description is too short. It is suggested to expand this section.
We agree that the results section is too less. I expanded this section.
- In line 62, a space is missing before “In contrast to”.
I have already a space in line 62, before “In contrast to”.
- In line 219, there is an additional comma in "(6 women, 7 men,)".
I have already removed the additional comma in "(6 women, 7 men,)", line 219.
- In line 380, please use italics for “C. trachomatis”.
I have already used italics for “C. trachomatis” in line 380.
- I personally believe that the specific results of 'Susceptibility assessments' can be plotted and included in the paper to make the data clearer.
|
Antibiotic susceptibility |
|||||||
|
Macrolides |
Tetracyclines |
Quinolones |
|||||
|
Phenotypic (MIC) |
Genotypic (n= 66) |
Phenotypic (MIC) |
Genotypic |
Genotypic |
|||
|
Azithromycin (mg/L) (n=10) |
T2611C |
A2057G |
A2058C |
A2059G |
Doxycycline (mg/L) (n=10) |
tet (C) gene (n=73) |
G248T (n=73) |
|
≤0.125 |
ND |
ND |
ND |
ND |
≤0.064 |
ND |
ND |
I have already included the results of 'Susceptibility assessments' in a table.
ND: not detected; MIC: Minimum inhibitory concentrations.
- I personally think the summary of the last paragraph can be separated into the fifth part of the article.
We agree with the reviewer that the summary of the last paragraph can be separated into the fifth part of the article. I have included the summary in the new section Conclusions.

Reviewer 2 Report
I have no any other comments
Author Response
Dear Reviewer,
Thank you very much for the comments and suggestions, I think they add great value to the article.